# Study protocol for a randomized controlled clinical trial of a multifaceted cognitive training program using video games in childhood cancer survivors

Carlos González-Pérez[1‡], Eduardo Fernández-Jiménez[2,3,4‡], Elena Moran[5], Helena Melero[6], Norberto Malpica[7], Juan Álvarez-Linera[8], Mario Alonso Puig[5], Diego Plaza[1], Antonio Pérez-Martínez[1,9,10*]

1 Department of Pediatric Hemato-Oncology, La Paz University Hospital, Madrid, Spain, 2 Department of Psychiatry, Clinical Psychology and Mental Health, La Paz University Hospital, Madrid, Spain, 3 Hospital La Paz Institute for Health Research, (IdiPAZ), Madrid, Spain, 4 Faculty of Law, Education and Humanities, Universidad Europea de Madrid, Villaviciosa de Odón, Madrid, Spain, 5 Juegaterapia Foundation, Madrid, Spain, 6 Department of Psychobiology and Methodology in Behavioral Sciences, Universidad Complutense de Madrid, Madrid, Spain, 7 Medical Image Analysis and Biometry Laboratory (LAIMBIO), Rey Juan Carlos University, Madrid, Spain, 8 Department of Radiodiagnosis, Ruber International Hospital, Madrid, Spain, 9 Pediatric Department, Autonomous University of Madrid, Madrid, Spain, 10 CIBERER-ISCIII. IdiPAZ-CNIO Pediatric Onco-Hematology Clinical Research Unit, Spanish National Cancer Research Center (CNIO), Madrid, Spain

‡ These authors contributed equally to this work and share first authorship.
* aperezmartinez@salud.madrid.org

## Abstract

This randomized controlled trial aims to evaluate the efficacy of a cognitive training program using video games in improving neuropsychological, neurological, immunological, and inflammatory parameters in childhood cancer survivors. This study will recruit 56 patients aged 8–17 years who have completed cancer treatment 1–8 years prior to enrollment. Participants will be randomized to either the video game intervention or waiting group. The primary objectives are analyzing potential changes in neuropsychological tests covering all neurocognitive domains, neuroimaging tests (structural, diffusion, and functional imaging), and immune and inflammatory biomarker levels after video game intervention. The secondary objectives are to define the prevalence of neurocognitive deficits in the study population, analyze psychological and emotional self-perception and parental perception after the intervention, and assess the feasibility of implementing this new intervention methodology. The inclusion criteria comprise specific diagnoses (central nervous system [CNS] cancer, hematologic malignancies, extracranial solid tumors, and nonmalignant hematological diseases requiring allogeneic hematopoietic progenitor transplantation) and treatments (CNS surgery, radiotherapy, intrathecal/intraventricular chemotherapy, neurotoxic systemic chemotherapy, and hematopoietic stem cell transplantation). Patients with active disease, relapse, or prior neurological or psychiatric pathology

**Data availability statement:** No datasets were generated or analysed during the current study. All relevant data from this study will be made available upon study completion.

**Funding:** This research was financed by private funds from the Juegaterapia Foundation (PI: APM, institutional code: PI-6221), which works with pediatric oncology patients through video games. There are no financial or personal relationships between researchers and companies that produce video games and consoles used in the neurocognitive training programs included in this study. The funders had no role in study design, data collection and analysis, decision to publish, or preparation of the manuscript.

**Competing interests:** The authors have declared that no competing interests exist.

will be excluded. This study will improve the understanding and management of neurocognitive sequelae in childhood cancer survivors and ultimately enhance their quality of life. Trial identifier: NCT06312969

---

## 1 Introduction

Childhood cancer is the leading cause of death from the disease in children in developed countries. Fortunately, advances in research have allowed better diagnosis and treatment in recent years, improving survival at 5 years to over 80% and to over 90% in some of the most frequent neoplasms, such as acute lymphoblastic leukemia (ALL) [1].

This rise in survival means that there is an increasing number of survivors of childhood cancer, reaching a prevalence between 450 and 1240 childhood cancer survivors per million people in Europe in the last 30 years [2]. This situation requires healthcare professionals and researchers to focus their attention not only on curing the disease but also on improving the quality of life of survivors.

Both the disease itself and the treatment administered can produce medium- and long-term sequelae that appear in more than half of the patients. Neurocognitive effects are among the most frequent and important. According to several studies, between 17 and 75% of cancer survivors present with this condition in variable grades between six months and 20 years after the end of treatment [3]. The most frequently affected areas are memory, learning, concentration, reasoning, executive functions, attention, and vision-spatial abilities [4]. This is a major problem affecting the quality of life of childhood cancer survivors and has been frequently neglected until now, with no established diagnostic and therapeutic strategies in pediatric hemato-oncology units.

In addition to neurocognitive involvement, there are some structural and functional changes in the central nervous system (CNS) that have been reported by neuroimaging in childhood cancer survivors. These changes have been observed even in children with tumors outside the CNS and those with normal neurocognitive capacity [5]. The most frequent structural alteration is the reduction in gray matter volume and changes in the integrity of white matter in different locations, especially in the frontal lobe [6–8]. Functional alterations are studied using functional magnetic resonance imaging (fMRI). Some studies have attempted to identify the brain areas underlying the neurocognitive effects of chemotherapy, proposing hypoactivation in the parietal and prefrontal areas as the leading cause [9]. These studies have been infrequent to date, especially in children, and have less clinical correlation.

Video games have been used for years as a training tool for executive functions in different pediatric pathologies such as attention-deficit/hyperactivity disorder (ADHD) [10,11]. Their potential is so important that in the United States the Food and Drug Administration (FDA) has approved a video game as a prescription to reduce the severity of ADHD in pediatric patients [12]. Other studies have demonstrated structural and functional changes in healthy people using neuroimaging techniques after the use of video games, with an increase in the activation of certain brain areas [13,14].

Although growing evidence supports the presence of neurocognitive sequelae in survivors of pediatric cancer, there is considerable variability in the type and severity

of deficits reported across studies, as well as in their estimated prevalence. Furthermore, the relationship between cognitive impairments and neurostructural or neurofunctional alterations remains unclear. Despite the increasing recognition of these late effects, there is no established routine for the systematic screening of neurocognitive sequelae in survivorship care, nor are there widely accepted, engaging, and effective cognitive training interventions tailored to this population. This study aims to address these gaps by proposing a multifaceted, game-based intervention and by exploring cognitive, neuroimaging, and immunological outcomes in an integrated manner.

## 2 Methods and analysis

### Study design

This project is a single-center, open-label, parallel group, two-armed randomized clinical trial evaluating changes in neurological, neuropsychological, immunological, and inflammatory parameters after a cognitive training program with video games compared to a waiting group without the program (Fig 1).

### Study objectives

**2.1.1 Primary objectives.** The primary objective of this trial is to demonstrate relevant changes in neurocognitive, neurological, immunological, and inflammatory evaluations after a training program with videogames. The primary endpoints are defined as follows:

- Clinically relevant improvement from moderate effect sizes in the following parameters measured by neuropsychological tests: Symbol Digit Modalities Test (SDMT), Continuous Performance Test, third edition (CPT 3), Spain-Complutense Verbal Learning Test for Children (TAVECI)/Spain-Complutense Verbal Learning Test (TAVEC), Vocabulary subtest of the Wechsler Intelligence Scale for Children, fifth edition (WISC-V)/ Weschler Intelligence Scale for adults, fourth edition (WAIS-IV) Vocabulary and Digit Span subtests, verbal fluency tests, STROOP, Rey-Osterrieth Complex Figure (ROCF), Test Of Non-verbal Intelligence, fourth edition (TONI-4) (forms A and B), and the Behavior Rating Inventory of Executive Function, second edition (BRIEF-2) questionnaire.

- Statistically significant changes in neuroimaging tests. The following variables will be measured:

  ◦ Structural imaging: volume measurement and Voxel Based Morphometry

  ◦ Diffusion Imaging: diffusion maps and structural connectivity

  ◦ Functional imaging: resting state and task based fMRI

- Statistically significant changes in immune and inflammatory biomarker levels before and after treatment:

  ◦ Study of lymphocyte populations by parametric flow cytometry: T lymphocytes, B lymphocytes, natural killer (NK) lymphocytes, NK T lymphocytes

  ◦ Study of inflammatory cytokines by LUMINEX: interleukin (IL) -2, IL-4, IL-6, Tumor necrosis factor (TNF) alpha, interferon (IFN) gamma, IL-10, IL-17TH, IL-1R antagonist

**2.1.2 Secondary objectives.**

- To define the prevalence of neurocognitive deficits in cancer survivors in our population.

- To analyze psychological and emotional self-perception and parent perception after controlled intervention with video games (Behavioral Assessment System for Children and Adolescents, third edition, BASC-3).

- To assess the feasibility of implementing a new intervention methodology through technological games.

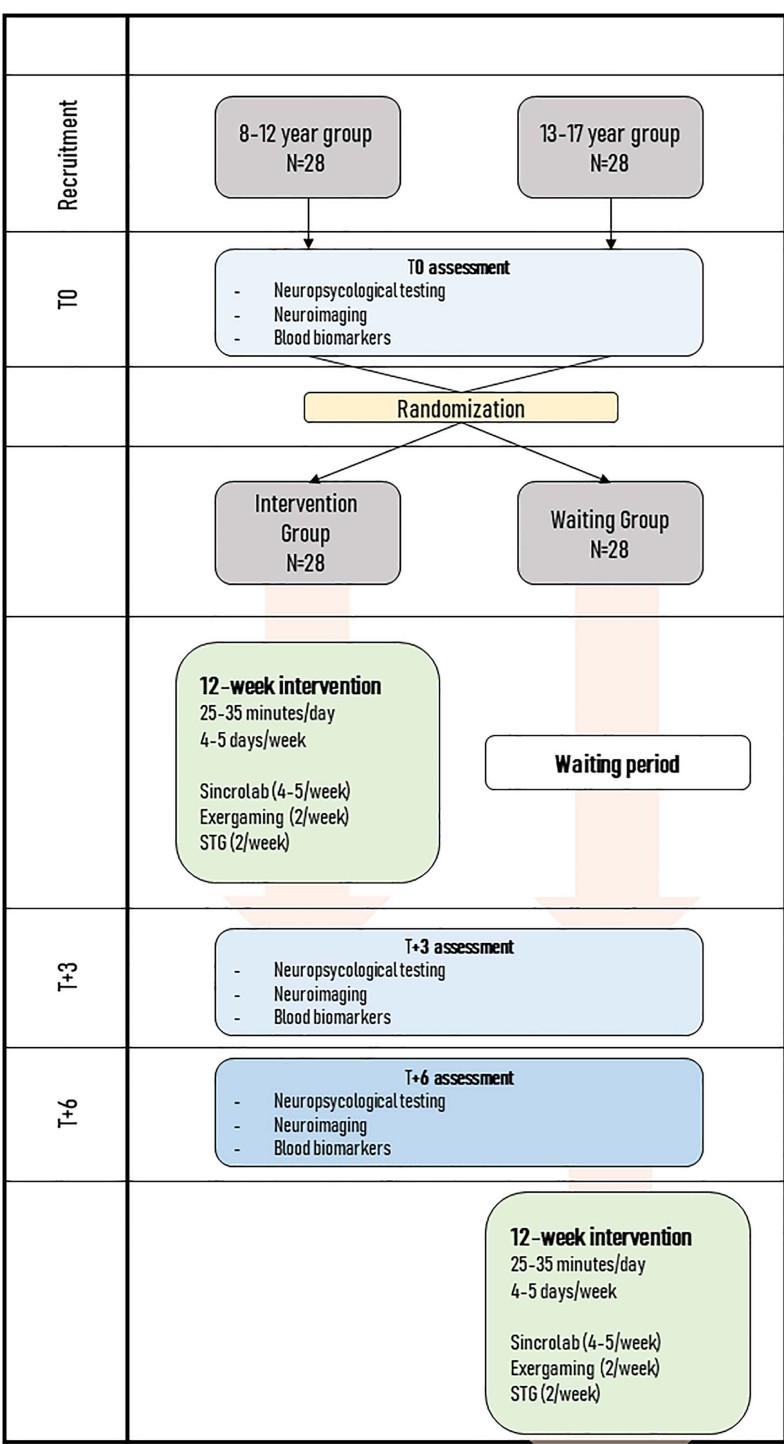

**Fig 1. Study algorithm.** Participants were divided into two age groups (8–12 and 13–17 years) and underwent baseline assessment (T0) at the time of recruitment. They were then randomized to either the intervention group or the waitlist control group. A second assessment (T + 3) was conducted 12 weeks later, followed by a third assessment (T + 6) 12 weeks later. After this period, the participants in the waitlist group were offered the opportunity to receive the intervention over the following 12 weeks. *STG = Skill Training Game.*

**Trial population**

A sample of 56 patients will be recruited in the study from among the patients of the Department of Pediatric Hemato-Oncology of an academic hospital in Spain between 8 and 17 years of age who have completed cancer treatment 1 to 8 years before enrolment. All patients provided written informed consent before any study procedure. The patients will be randomized in a 1:1 ratio to either the treatment group or the waiting group, after stratification for age, one group from 8 to 12 years, and the other group from 13 to 17 years of age.

There will also be a control group of healthy patients (who have not suffered an oncological disease) in which structural and functional MRI will be performed with the same protocol as in the waiting and treatment groups. This group will establish a standard of comparison for neuroimaging tests of patients in the study. It is intended to recruit 20 patients in this group, 10 patients aged 8 to 12 years, and 10 patients aged 13 to 17 years. Patients will be selected from among siblings or other families of the intervention/waiting group.

Recruitment will be made from the patients of the Department of Pediatric Hemato-Oncology follow-up visits via telephone calls and face-to-face appointments if required to explain the study to both parents and participants themselves.

### 2.1.3 Selection criteria for the treatment/waiting group.

**Inclusion criteria**

- Patients aged between 8 and 17 years at the time of recruitment.

- Have completed treatment between one and eight years prior to enrollment.

- Have had one of the following diagnoses:

  ◦ CNS cancer (posterior fossa tumors and supratentorial gliomas < 1 cm affecting the associative areas).

  ◦ Hematologic malignancies (leukemia or lymphoma).

  ◦ Extracranial solid tumors.

  ◦ Nonmalignant hematological diseases and indications for allogeneic hematopoietic progenitor transplantation.

- Having received at least one of the following treatments:

  ◦ CNS surgery.

  ◦ CNS radiotherapy.

  ◦ Intrathecal and intraventricular chemotherapy.

  ◦ Neurotoxic systemic chemotherapy.

  ◦ Hematopoietic stem cell transplantation.

- Informed consent was obtained from the patient's parent/guardian.

**Exclusion criteria**

- Active oncologic disease or relapse.

- Prior neurological or psychiatric pathology that may preclude trial or treatment evaluations:

  ◦ Psychological or neurocognitive illness or sequelae that preclude neuropsychological assessment or are expected to significantly artifact MRI results (e.g., significant decrease in visual acuity, CNS surgical scar that artifacts imaging results, severe cognitive delay that precludes testing).

  ◦ Psychological or neurocognitive illnesses or sequelae that prevent or contraindicate the use of video games (epilepsy that prevents the use of screens, significant decrease in visual acuity, etc.).

- Mild or self-limiting neurological or psychiatric pathology that does not interfere with trial diagnosis and treatment (headache, epilepsy in remission with effective treatment, mild cognitive delay) will be allowed.

- Current or recent (less than one year) use of other cognitive stimulation or brain training that may interfere with the study results.

- Refusal to abstain from the use of the study treatment games in case of being assigned to Group B (waiting group).

- Medical treatment that may significantly interfere with neuropsychological, imaging or biomarker assessments.

### 2.1.4 Selection criteria for the control group of healthy patients.

**Inclusion criteria**

- Patients aged between 8 and 17 years at the time of recruitment.

- Informed consent was obtained from the patient's parent/guardian.

**Exclusion criteria**

- History of oncological disease and/or treatment with neurotoxic potential.

- Psychological or neurocognitive illness or sequelae that preclude neuropsychological assessment or are expected to significantly artifact MRI results (e.g., significant decrease in visual acuity, CNS surgical scar that artifacts imaging results, severe cognitive delay that precludes testing).

- Current or recent (less than one year) use of other cognitive stimulation or brain training that may interfere with the study results.

### 2.1.5 Study withdrawal.

- Death.

- Relapse of the underlying disease or other oncological pathologies or pathologies requiring intensive medical treatment.

- Decision of the patient or their parent/guardian.

- Concomitant use of another cognitive training platform.

- Use of experimental group games among patients in group B (control group).

- Medical indication for treatment that interferes with trial measures. Mildly interfering treatments (e.g., methylphenidate-derived drugs) will be allowed, provided that the date of treatment initiation, date of dose modification, and dosages are documented.

**Variables and instruments**

The variables measured in this project, the instruments used and the informants who will complete each measure are detailed below:

- Sociodemographic (sex, age, educational level) and clinical variables (weight and height, blood pressure, heart rate, physical examination, and neurological examination). This data will be collected with an initial survey created *ad hoc* for the present study.

- Ecological Executive Functioning. The BRIEF-2 questionnaire [15] will be used. This questionnaire will be completed by the patient's parents or legal guardians.

- Emotional and behavioral problems. The BASC-3 [16] will be administered, including both parent-reported and self-reported versions. This questionnaire assesses the personal and interpersonal/social domains of each patient, covering areas such as adaptive skills, resiliency, externalizing and internalizing problems, school problems, and relationships with parents and peers.

- Cognitive processing speed. The SDMT [17] will be administered to patients, in its oral form, to assess their processing speed of information.

- Visual sustained attention/vigilance. The CPT 3 [18], will be administered to the patients. This test computes several scores, as follows: Detectability, Omissions, Commissions, Perseverations, Hit Reaction Time (HRT), HRT Standard Deviation, Variability, HRT Block Change and HRT Inter-Stimulus Interval Change.

- Immediate and delayed verbal memory. A Spanish test based on the California Verbal Learning Test will be used. Specifically, the TAVECI [19] will be administered to patients aged 8–16 years, and the TAVEC version [20] will be applied to patients aged 16–18 years. This test allows for the calculation of a wide range of scores, and in this study, the following scores will be used: Immediate Recall, Short Delay Free Recall, Short Delay Cued Recall, Long Delay Free Recall, Long Delay Cued Recall, Long Delay Recognition, Semantic clustering strategies, Serial clustering strategies, False positives, and Discriminability.

- Semantic memory. The WISC-V [21] will be administered to patients up to the age of 16.11 years, and the adult version, fourth edition (WAIS-IV) [22], will be administered to patients up to 18 years of age.

- Verbal working memory and cognitive flexibility. The Digit Span subtest of the WISC-V [21] or WAIS-IV version [22] will be used. Three neurocognitive domain-specific scores will be considered (Forward, Backward, and Sequencing subscales), as well as the global score (Digit Span), as a measure of cognitive flexibility [21].

- Verbal fluency. The Verbal Fluency Test (TFV) [23] will be administered to patients, calculating the following eight scores: Phonological fluency, Semantic fluency, Excluded-letter fluency, Total Fluency Index, Errors in phonological fluency, Errors in semantic fluency, Errors in excluded-letter fluency and Total errors.

- Cognitive interference inhibition. The STROOP test [24] will be applied to patients and the following four scores will be computed: the Word reading, the Color naming, the Color-word and Resistance to Interference.

- Visuo-constructive organizational ability and immediate and delayed visual memory. The Rey Complex Figure Test (RCFT) [25] will be applied to patients, generating the following four scores: a copy score (accuracy regarding visual–spatial constructional ability), time required to copy the figure, and two delayed recall scores (at 3 and 30 min).

- Non-verbal Intelligence. Both forms (A and B) of the TONI-4 [26] will be administered to the patients.

- Structural image: volume of white matter, gray matter, total intracranial volume, volume of cerebral lobes, and different subcortical structures. Voxel-based morphometry (VBM).

- Diffusion Imaging: diffusion maps (fractional anisotropy, mean diffusivity, etc.) and structural integrity.

- Functional imaging: Task-based fMRI: functional differences related to a working memory task; Resting State fMRI: analysis of functional connectivity (Seed Based Analysis and Independent Component Analysis).

- Study of lymphocyte populations by parametric flow cytometry: T lymphocytes, B lymphocytes, NK lymphocytes, and NK T lymphocytes.

- Study of inflammatory cytokines: IL-2, IL-4, IL-6, TNF alpha, IFN gamma, IL-10, IL-17a, IL-1R antagonist.

All neuropsychological tests and patient questionnaires will be administered by clinical neuropsychologists from the Department of Psychiatry, Clinical Psychology and Mental Health of the hospital. Neuroimaging tests will be performed by trained technicians, and the data will be analyzed by researchers with expertise in structural and functional neuroimaging.

The selected inflammatory and immune biomarkers were chosen based on prior evidence of their involvement in cancer-related cognitive impairment (CRCI) and neuroinflammation. Pro-inflammatory cytokines such as IL-6 and TNF-α have been associated with cognitive dysfunction in both pediatric and adult oncology populations. IL-10 and IL-1 receptor antagonists (IL-1RA) represent anti-inflammatory responses that may indicate compensatory immune regulation. The inclusion of Th1 (e.g., IL-2, IFN-γ), Th2 (e.g., IL-4, IL-10), and Th17 (e.g., IL-17a) cytokines provides a broader immunological context to explore the potential links between immune activation profiles and cognitive or neural changes in survivors.

### Interventions

**2.1.6 Experimental intervention.** The patient will receive treatment for a period of 12 weeks, in which they will commit to using the video games of the intervention with the following pattern:

- "Brain-training game" (Sincrolab®): sessions of 7-12 minutes with a frequency of 4 days a week.

- "Exer-gaming" (Just Dance® in the 8-12 year-old group and Ring Fit Adventure® in the 13-17 year-old group): sessions of 15-20 minutes 2 days a week.

- "Skill-training games" (Minecraft Education®): sessions of 15-20 minutes 2 days a week.

Adherence to the training program will be monitored using the game's own game-tracking system (Sincrolab and Minecraft Education). In the case of games without their own system (the two exer-gaming games), video calls will be made with the participant during game time, in addition to weekly phone calls, to ensure the correct follow-up of the program, resolve doubts, and provide technical support.

**2.1.7 Waiting group.** These patients will not receive the experimental training program for the duration of the trial and will only be evaluated as the intervention group. Nonetheless, every patient will be offered the training program after completing the study.

### Informed consent procedure

Informed consent will be obtained before any specific study procedure is performed. Eligible subjects and their family members or legal representative(s) will be informed about the objectives and procedures before the start of the study. Minors older than 12 years of age, in addition to the informed consent signed by their legal representative, must give their consent.

### Schedule of trial procedures

Table 1 summarizes the procedures that will be performed for each patient.

### Statistical methods

The data will be processed primarily using the statistical software IBM SPSS Statistics 30.0, among others, and the following analyses will be performed:

Initially, descriptive analyses will be performed on all the variables to detect any potential errors or outliers in the dataset. Zero-order correlation analyses will then be computed to determine the strength of association between the variables, and partial correlation analyses will be conducted to assess the association while excluding the influence of confounding variables.

The main statistical analysis will consist of a mixed Analysis of Covariance (ANCOVA) to evaluate between-group effects (intervention vs. waiting list), within-group effects over time (T0, T+3, and T+6), and their interactions, controlling for relevant covariates. These covariates will include age, sex, baseline cognitive performance, cancer diagnosis, and treatment modality

**Table 1. Schedule of Trial Procedures.**

| Trial period | Screening | T0 | Treatment/waiting period | | | | | | T+3 | T+6 |
|---|---|---|---|---|---|---|---|---|---|---|
| Weeks | −4 | −4 to −1 | 1 | 3 | 5 | 7 | 9 | 11 | 13 to 17 | 25 to 29 |
| Informed consent | X | | | | | | | | | |
| Inclusion and exclusion criteria | X | | | | | | | | | |
| Demographic data and medical history | X | | | | | | | | | |
| Concomitant medication | X | | | | | X | | | X | |
| Incidents during treatment | X | | | | | X | | | X | |
| Physical examination and vital signs | X | | | | | | | | X | |
| Biomarkers | | X | | | | | | | X | X |
| Neuropsychological evaluation | | X | | | | | | | X | X |
| Neuroimaging evaluation | | X | | | | | | | X | X |
| Habits and psychosocial survey | | X | | | | | | | | X |
| Adherence evaluation* | | | X | X | X | X | X | X | | |

*Only in the intervention group. Study investigators will review the activity recorded by compatible gaming platforms, and will contact the patient and family to verify adherence to treatment.

(e.g., CNS radiation, neurotoxic chemotherapy). *P*-value adjustments will be performed for multiple comparisons. Assumptions of normality, homogeneity of variances, and sphericity will be examined and addressed as needed. Additional analyses will include latent growth models to examine cognitive trajectories, and multiple linear regression models to determine the relative contribution of predictor variables on outcome measures, as previously described [27].

The diagnostic group (CNS vs. non-CNS tumors) will be included as a covariate in the main analyses, and exploratory subgroup comparisons between CNS and non-CNS survivors will be conducted to assess the potential differences in cognitive, neuroimaging, and biomarker outcomes.

All statistical analyses will be complemented with appropriate effect size indices for each statistical test (e.g., $f$, omega square, and $R^2$ indices) [28]. Additionally, the Reliable Change Index (RCI) will be computed to determine the real change in each individual score between measurements over time [29].

The original power analysis was performed using GPower 3.1.9.4, assuming a repeated-measures ANCOVA ($2 \times 3$ design) with a statistical power of 0.95, $\alpha = 0.05$, and a moderate effect size ($f = 0.25$), resulting in a minimum required sample size of 36 participants. However, because GPower does not account for intra-subject correlation (ICC), this estimation corresponds to a fixed-effects model. As the analysis will use a mixed model with repeated measures, we acknowledge that the ICC could affect power. To increase transparency, we have considered ICC values between 0.3 and 0.6, which suggest that between 42 and 48 participants may be needed to maintain the same level of power. This is feasible within our recruitment plan.

Missing data will be handled using multiple imputation under the missing-at-random (MAR) assumption. Sensitivity analyses will compare the imputed results with complete-case analyses, and other techniques, such as pattern-mixture models, may be used if appropriate.Randomization will be performed automatically using the REDCap platform and stratified by age group. Once a patient has been assigned, the study investigators will inform them and their families via telephone calls.

Investigators responsible for the clinical follow-up will not be blinded. However, those conducting neuropsychological and neuroimaging assessments will be blinded to group allocation to minimize bias in the outcome evaluation.

## Monitoring

The study will be monitored by the Central Unit for Clinical Research and Clinical Trials of La Paz University Hospital (UCICEC-HULP), which is independent of the sponsors and investigators. No adverse effects are expected from the

assessment tests or treatment. Monitors will review the data collected by the study investigators and report any errors and deviations from the protocol.

## Status of the study

Recruitment started in February 2023, and is expected to be completed by February 2025. Data collection will be completed by September 2025, and the results will be expected by the end of 2025.

## Ethics and dissemination

This study was reviewed and approved by the Ethics Committee of La Paz University Hospital in Madrid (approval date: 2022-08-04, institutional code: PI-6221). The study protocol was registered at clinicaltrials.gov (NCT06312969). The patients provided written informed consent to participate in the study. This study was designed in accordance with the SPIRIT and CONSORT guidelines (Fig 2).

Participants' personal information will always be anonymized and stored in encrypted databases, and will only be known to the study investigators.

The investigators will publish all the findings of the study, as well as the study protocol, and communicate important protocol modifications. Authorship criteria will follow the CRediT (Contributor Roles Taxonomy) guidelines.Upon study completion and publication of the main findings, de-identified raw datasets, including neuroimaging and biomarker data, will be made available upon reasonable request to qualified researchers and subject to approval by the corresponding ethics committee. This ensures transparency while protecting participant confidentiality.

## 3 Discussion

The ultimate goal of this study is to design new diagnostic and treatment strategies for neurological and neurocognitive sequelae in pediatric cancer patients. These sequelae are often neglected in pediatric hemato-oncology units, as no systematic screening and treatment plans have been established. This generates discomfort and a decrease in the quality of life of both patients and their families, who also report feeling helpless.

The number of surviving childhood cancer patients is increasing, owing to advances in research that allow for better diagnosis and management of the disease [30]. Therefore, in recent years, it has become clear that there is a need to create specialized teams for the follow-up of survivors within pediatric hemato-oncology units, or even in specific centers dedicated to this purpose [31]. New strategies for better diagnosis and treatment of the long-term effects of the disease are urgently needed.

The cognitive impairment experienced by cancer survivors requires special attention. This clinical condition has received different names, from "chemo-brain" or chemotherapy-induced cognitive impairment (CICI) [32] to cancer-related cognitive impairment (CRCI), given that some patients experience these deficits even before the onset of treatment [33]. Although there are multiple studies and reviews on this disorder, there are few well-established neurocognitive care programs for surviving patients in most pediatric haemato-oncology units.

Video games have been used for various purposes in pediatric oncology. It is well known that play has a therapeutic role in these patients and is a very important element in children's neurodevelopment, especially in situations of adversity or stress [34], having an impact on their social interactions, identity development, and communication [35]. It has been observed that the introduction of video games into routine clinical practice minimizes procedural pain and anxiety for both pediatric patients and their caregivers [36,37].

Programs called "Child Life Services" have emerged recently, formed by professionals who guide patients and their families through the disease process (diagnosis, complications, hospitalizations, etc.), helping them to process and cope with stressful situations using play as a fundamental tool [38].

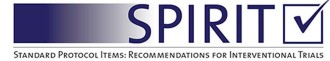

SPIRIT ☑
STANDARD PROTOCOL ITEMS: RECOMMENDATIONS FOR INTERVENTIONAL TRIALS

SPIRIT 2013 Checklist: Recommended items to address in a clinical trial protocol and related documents*

| Section/item | Item No | Description | Addressed on page number |
|---|---|---|---|
| **Administrative information** | | | |
| Title | 1 | Descriptive title identifying the study design, population, interventions, and, if applicable, trial acronym | 1 |
| Trial registration | 2a | Trial identifier and registry name. If not yet registered, name of intended registry | 1 |
| | 2b | All items from the World Health Organization Trial Registration Data Set | Throughout the manuscript |
| Protocol version | 3 | Date and version identifier | 1 |
| Funding | 4 | Sources and types of financial, material, and other support | 13 |
| Roles and responsibilities | 5a | Names, affiliations, and roles of protocol contributors | 1 |
| | 5b | Name and contact information for the trial sponsor | 1 |
| | 5c | Role of study sponsor and funders, if any, in study design; collection, management, analysis, and interpretation of data; writing of the report; and the decision to submit the report for publication, including whether they will have ultimate authority over any of these activities | 13 |
| | 5d | Composition, roles, and responsibilities of the coordinating centre, steering committee, endpoint adjudication committee, data management team, and other individuals or groups overseeing the trial, if applicable (see Item 21a for data monitoring committee) | 10 |
| **Introduction** | | | |
| Background and rationale | 6a | Description of research question and justification for undertaking the trial, including summary of relevant studies (published and unpublished) examining benefits and harms for each intervention | 2, 3 |
| | 6b | Explanation for choice of comparators | N/A |
| Objectives | 7 | Specific objectives or hypotheses | 5 |
| Trial design | 8 | Description of trial design including type of trial (eg, parallel group, crossover, factorial, single group), allocation ratio, and framework (eg, superiority, equivalence, noninferiority, exploratory) | 3 |
| **Methods: Participants, interventions, and outcomes** | | | |
| Study setting | 9 | Description of study settings (eg, community clinic, academic hospital) and list of countries where data will be collected. Reference to where list of study sites can be obtained | 5 |
| Eligibility criteria | 10 | Inclusion and exclusion criteria for participants. If applicable, eligibility criteria for study centres and individuals who will perform the interventions (eg, surgeons, psychotherapists) | 6 |
| Interventions | 11a | Interventions for each group with sufficient detail to allow replication, including how and when they will be administered | 8 |
| | 11b | Criteria for discontinuing or modifying allocated interventions for a given trial participant (eg, drug dose change in response to harms, participant request, or improving/worsening disease) | 6, 7 |
| | 11c | Strategies to improve adherence to intervention protocols, and any procedures for monitoring adherence (eg, drug tablet return, laboratory tests) | 8 |
| | 11d | Relevant concomitant care and interventions that are permitted or prohibited during the trial | 6, 7 |
| Outcomes | 12 | Primary, secondary, and other outcomes, including the specific measurement variable (eg, systolic blood pressure), analysis metric (eg, change from baseline, final value, time to event), method of aggregation (eg, median, proportion), and time point for each outcome. Explanation of the clinical relevance of chosen efficacy and harm outcomes is strongly recommended | 7, 8 |
| Participant timeline | 13 | Time schedule of enrolment, interventions (including any run-ins and washouts), assessments, and visits for participants. A schematic diagram is highly recommended (see Figure) | 9 |
| Sample size | 14 | Estimated number of participants needed to achieve study objectives and how it was determined, including clinical and statistical assumptions supporting any sample size calculations | 5, 10 |
| Recruitment | 15 | Strategies for achieving adequate participant enrolment to reach target sample size | 6 |
| **Methods: Assignment of interventions (for controlled trials)** | | | |
| Allocation: | | | |
| Sequence generation | 16a | Method of generating the allocation sequence (eg, computer-generated random numbers), and list of any factors for stratification. To reduce predictability of a random sequence, details of any planned restriction (eg, blocking) should be provided in a separate document that is unavailable to those who enrol participants or assign interventions | 10 |
| Allocation concealment mechanism | 16b | Mechanism of implementing the allocation sequence (eg, central telephone; sequentially numbered, opaque, sealed envelopes), describing any steps to conceal the sequence until interventions are assigned | 10 |
| Implementation | 16c | Who will generate the allocation sequence, who will enrol participants, and who will assign participants to interventions | 10 |
| Blinding (masking) | 17a | Who will be blinded after assignment to interventions (eg, trial participants, care providers, outcome assessors, data analysts), and how | 10 |
| | 17b | If blinded, circumstances under which unblinding is permissible, and procedure for revealing a participant's allocated intervention during the trial | 10 |
| **Methods: Data collection, management, and analysis** | | | |
| Data collection methods | 18a | Plans for assessment and collection of outcome, baseline, and other trial data, including any related processes to promote data quality (eg, duplicate measurements, training of assessors) and a description of study instruments (eg, questionnaires, laboratory tests) along with their reliability and validity, if known. Reference to where data collection forms can be found, if not in the protocol | 7, 8 |
| | 18b | Plans to promote participant retention and complete follow-up, including list of any outcome data to be collected for participants who discontinue or deviate from intervention protocols | 7, 8 |

| | | | |
|---|---|---|---|
| Data management | 19 | Plans for data entry, coding, security, and storage, including any related processes to promote data quality (eg, double data entry; range checks for data values). Reference to where details of data management procedures can be found, if not in the protocol | 12 |
| Statistical methods | 20a | Statistical methods for analysing primary and secondary outcomes. Reference to where other details of the statistical analysis plan can be found, if not in the protocol | 10 |
| | 20b | Methods for any additional analyses (eg, subgroup and adjusted analyses) | 10 |
| | 20c | Definition of analysis population relating to protocol non-adherence (eg, as randomised analysis), and any statistical methods to handle missing data (eg, multiple imputation) | 10 |
| **Methods: Monitoring** | | | |
| Data monitoring | 21a | Composition of data monitoring committee (DMC); summary of its role and reporting structure; statement of whether it is independent from the sponsor and competing interests; and reference to where further details about its charter can be found, if not in the protocol. Alternatively, an explanation of why a DMC is not needed | 10 |
| | 21b | Description of any interim analyses and stopping guidelines, including who will have access to these interim results and make the final decision to terminate the trial | 10 |
| Harms | 22 | Plans for collecting, assessing, reporting, and managing solicited and spontaneously reported adverse events and other unintended effects of trial interventions or trial conduct | 10 |
| Auditing | 23 | Frequency and procedures for auditing trial conduct, if any, and whether the process will be independent from investigators and the sponsor | 10 |
| **Ethics and dissemination** | | | |
| Research ethics approval | 24 | Plans for seeking research ethics committee/institutional review board (REC/IRB) approval | 12 |
| Protocol amendments | 25 | Plans for communicating important protocol modifications (eg, changes to eligibility criteria, outcomes, analyses) to relevant parties (eg, investigators, REC/IRBs, trial participants, trial registries, journals, regulators) | 13 |
| Consent or assent | 26a | Who will obtain informed consent or assent from potential trial participants or authorised surrogates, and how (see Item 32) | 12 |
| | 26b | Additional consent provisions for collection and use of participant data and biological specimens in ancillary studies, if applicable | N/A |
| Confidentiality | 27 | How personal information about potential and enrolled participants will be collected, shared, and maintained in order to protect confidentiality before, during, and after the trial | 12 |
| Declaration of interests | 28 | Financial and other competing interests for principal investigators for the overall trial and each study site | 13 |
| Access to data | 29 | Statement of who will have access to the final trial dataset, and disclosure of contractual agreements that limit such access for investigators | 12 |
| Ancillary and post-trial care | 30 | Provisions, if any, for ancillary and post-trial care, and for compensation to those who suffer harm from trial participation | N/A |
| Dissemination policy | 31a | Plans for investigators and sponsor to communicate trial results to participants, healthcare professionals, the public, and other relevant groups (eg, via publication, reporting in results databases, or other data sharing arrangements), including any publication restrictions | 13 |
| | 31b | Authorship eligibility guidelines and any intended use of professional writers | 13 |
| | 31c | Plans, if any, for granting public access to the full protocol, participant-level dataset, and statistical code | 13 |
| **Appendices** | | | |
| Informed consent materials | 32 | Model consent form and other related documentation given to participants and authorised surrogates | Appendices |
| Biological specimens | 33 | Plans for collection, laboratory evaluation, and storage of biological specimens for genetic or molecular analysis in the current trial and for future use in ancillary studies, if applicable | 8 |

*It is strongly recommended that this checklist be read in conjunction with the SPIRIT 2013 Explanation & Elaboration for important clarification on the items. Amendments to the protocol should be tracked and dated. The SPIRIT checklist is copyrighted by the SPIRIT Group under the Creative Commons "Attribution-NonCommercial-NoDerivs 3.0 Unported" license.

**Fig 2. Checklist.** SPIRIT Guidelines.

In recent years, there has been an increase in studies using video games for cognitive training with good results, especially in patients with ADHD, which has improved attentional outcomes [39].

Because of these results, it has also been implemented in cancer survivors with CRCI. Two main groups have conducted studies on this pediatric population. The first was the St Jude's Children's Research Hospital group [40], which employed a computerized cognitive intervention (COGMED) in 68 patients surviving acute lymphoblastic leukemia (ALL) or brain tumors (BT). They demonstrated that their intervention was feasible and well accepted by their patients [41]. Their results showed an improvement over the control group in working memory, attention, and processing speed, as well as a reduction in activation of the left lateral prefrontal and bilateral medial frontal areas on fMRI [40]. A more recent study revealed that the neurocognitive effects of the intervention were maintained for up to six months later [42]. The second group was from Bern, Switzerland [43]. They designed a study involving 69 childhood cancer survivors (both with CNS and non-CNS involvement, with previous treatment with chemotherapy and/or radiotherapy), comparing three groups: a control group, a cognitive training group (COGMED), and an exergaming group (Xbox Kinect) [43]. They reported an improvement in visual working memory in patients in the COGMED arm [44].

Independently, another research group conducted a study with a smaller sample size (20 surviving ALL patients) using COGMED in patients with reported attention or working memory deficits, and also observed improvements in visual working memory and parent-reported language problems [45].

All of these studies showed promising results in the treatment of neurocognitive sequelae in pediatric oncology survivors through the use of video games. However, there is a paucity of randomized clinical trials on multifaceted cognitive training programs that combine brain-training games, physical exercise (exer-gaming), and commercial games that allow the development of certain skills, such as visuospatial skills and memory. We hope that this combination of games is a treatment with good compliance by patients, that it is attractive to them, and that it improves their neurocognitive profile in different areas.

Unlike previous investigations, the present research project employs three different types of endpoints: neurocognitive assessment, neuroimaging, and biomarkers. Alterations in neuropsychological areas, such as working memory or attention, have been reported previously, while there is less information about other areas, such as language or visuospatial ability. Therefore, the neuropsychological assessment of this project, designed specifically for the evaluation of childhood cancer survivors, encompasses multiple areas and allows for a more global view of neurocognitive deficits among these patients.

In addition, some structural and functional changes in MRI have been proposed as the basis for certain deficits in specific cognitive areas; however, these associations have not been demonstrated. By studying childhood cancer survivors, we hope to find common patterns or profiles of structural and functional involvement that may correspond to alterations in cognitive function detected by neurocognitive testing. The observed patterns may reflect compensatory mechanisms or treatment-related alterations, consistent with previous reports in pediatric populations.

This study also incorporates immunological and inflammatory markers that may be altered after cancer treatment, which may play a role in CRCI. Some studies have suggested a decrease in inflammatory parameters in patients with the use of certain video games [46], but this variable has not been incorporated in most trials prior to this study. If confirmed, these findings could support a link between systemic inflammation and cognitive impairment, highlighting new targets for early intervention.

However, the interpretation of inflammatory biomarkers such as IL-6 or TNF-α, must be approached with caution, given their well-documented biological variability influenced by circadian rhythms, stress, infections, or individual immune profiles. To minimize this variability, cytokine sampling in this study will be conducted under standardized conditions and processed centrally using validated multiplex immunoassays. Although no universal clinical cutoff values exist for these markers in pediatric cancer survivors, comparisons will be based on within-subject changes, between-group differences, and reference ranges from healthy controls. Our approach prioritizes the detection of patterns of biological modulation associated with cognitive changes, even in the absence of pre-specified clinical thresholds.

This study has several limitations. The main limitation is the heterogeneity of the sample due to the wide age range regarding some clinical outcomes and the performance of the interventions, and the diversity of diagnoses and treatments received. This, together with the limited sample size, could decrease the statistical power of the study.

Another possible limitation is the lack of comparison standards for certain variables. While neuropsychological tests have sex- and/or age-adjusted norms, functional neuroimaging tests are not frequent in pediatrics, so there are no large series in healthy children with which to compare. Therefore, we added a control group of healthy patients to allow us to establish a normal pattern by age.

In addition, the inclusion criterion of selecting survivors who are 1–8 years post-treatment may introduce survivor bias, as it excludes patients with early relapse or severe sequelae who may not be able to participate. As a result, the findings may not be generalizable to all pediatric cancer survivors, but rather to those with a relatively stable follow-up.

Future studies should explore the long-term sustainability of cognitive improvements following this type of intervention, and examine whether earlier implementation, closer to the end of treatment, may enhance its effectiveness. Further research is needed to validate the relationship between changes in cognitive function and neuroimaging or biomarker profiles in larger multicenter cohorts.

## 4 Conclusions

Increased survival in pediatric cancer patients highlights the urgent need to diagnose and treat the neuropsychological effects of both the disease and its treatments. Although this condition has been widely studied, we do not have standardized screening protocols or evidence-based cognitive rehabilitation tools currently available in routine clinical practice. By evaluating neuropsychological performance, neuroimaging variables, and blood biomarkers, this study will help determine the prevalence, severity and potential mechanisms of neurocognitive impairment in childhood cancer survivors. In parallel, it will assess the feasibility and effectiveness of a multifaceted cognitive training platform using video games, which -if beneficial- could offer a novel, engaging and scalable strategy for improving long-term outcomes in this population.

## Acknowledgments

We gratefully acknowledge the contributions of other researchers to the design of this project.

## Author contributions

**Conceptualization:** Carlos González-Pérez, Eduardo Fernández-Jiménez, Elena Moran, Norberto Malpica, Mario Alonso Puig, Antonio Pérez-Martínez.

**Data curation:** Carlos González-Pérez, Eduardo Fernández-Jiménez, Helena Melero.

**Formal analysis:** Eduardo Fernández-Jiménez, Helena Melero, Norberto Malpica.

**Funding acquisition:** Carlos González-Pérez, Mario Alonso Puig, Antonio Pérez-Martínez.

**Investigation:** Carlos González-Pérez, Eduardo Fernández-Jiménez, Elena Moran, Helena Melero, Norberto Malpica, Juan Álvarez-Linera, Diego Plaza, Antonio Pérez-Martínez.

**Methodology:** Carlos González-Pérez, Eduardo Fernández-Jiménez, Elena Moran, Helena Melero, Norberto Malpica, Juan Álvarez-Linera, Antonio Pérez-Martínez.

**Project administration:** Carlos González-Pérez, Eduardo Fernández-Jiménez, Antonio Pérez-Martínez.

**Resources:** Carlos González-Pérez, Mario Alonso Puig, Antonio Pérez-Martínez.

**Software:** Helena Melero, Norberto Malpica, Juan Álvarez-Linera.

**Supervision:** Carlos González-Pérez, Eduardo Fernández-Jiménez, Elena Moran, Mario Alonso Puig, Antonio Pérez-Martínez.

**Writing – original draft:** Carlos González-Pérez, Eduardo Fernández-Jiménez, Antonio Pérez-Martínez.

**Writing – review & editing:** Carlos González-Pérez, Eduardo Fernández-Jiménez, Elena Moran, Helena Melero, Norberto Malpica, Juan Álvarez-Linera, Mario Alonso Puig, Diego Plaza, Antonio Pérez-Martínez.

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
