## [Decision Letter · Decision Letter 0]

25 Apr 2025

Dear Dr. González-Pérez,

We look forward to receiving your revised manuscript.

Kind regards,

Miray Budak

Academic Editor

PLOS ONE

Journal Requirements:

4. We note that the original protocol that you have uploaded as a Supporting Information file contains an institutional logo. As this logo is likely copyrighted, we ask that you please remove it from this file and upload an updated version upon resubmission.

Reviewers' comments:

Reviewer's Responses to Questions

Comments to the Author

1. Does the manuscript provide a valid rationale for the proposed study, with clearly identified and justified research questions?

Reviewer #1: Yes

Reviewer #2: Yes

Reviewer #3: Partly

2. Is the protocol technically sound and planned in a manner that will lead to a meaningful outcome and allow testing the stated hypotheses?

Reviewer #1: Yes

Reviewer #2: Yes

Reviewer #3: Yes

3. Is the methodology feasible and described in sufficient detail to allow the work to be replicable?

Reviewer #1: No

Reviewer #2: Yes

Reviewer #3: Yes

4. Have the authors described where all data underlying the findings will be made available when the study is complete?

Reviewer #1: No

Reviewer #2: Yes

Reviewer #3: Yes

5. Is the manuscript presented in an intelligible fashion and written in standard English?

Reviewer #1: Yes

Reviewer #2: Yes

Reviewer #3: Yes

You may also provide optional suggestions and comments to authors that they might find helpful in planning their study.

Reviewer #1: The manuscript titled "Randomized controlled clinical trial with a multifaceted cognitive training program using video games in childhood cancer survivors" presents a well-designed randomized controlled trial (RCT) investigating an intervention for neurocognitive deficits. While the study is innovative and addresses an important clinical issue, several methodological and statistical issues require clarification to strengthen the manuscript.

Major Comments

1. Sample Size Justification and Power Calculation

The manuscript states that power was calculated using a mixed ANCOVA (2 × 3 design) assuming: Statistical power: 0.95, Alpha: 0.05, Effect size: f = 0.25 (moderate), Software: G*Power 3.1.9.4, Minimum sample size: 36 patients

While this information is useful, several key concerns remain:

Fixed-Effects vs. Mixed-Effects Calculation

o G*Power does not support true mixed-effects models, suggesting that the power calculation was based on a fixed-effects ANCOVA.

o However, the study actually employs a mixed-effects model with repeated measures, which accounts for within-subject correlations.

Please discuss Potential Impact of the Model Choice on Power. Should intra-subject correlation (ICC) be properly accounted for in the sample size calculation? The authors should clarify whether intra-subject correlation was assumed in power estimation and consider reporting different ICC scenarios to improve transparency including whether the shift from fixed-effects ANCOVA to a mixed model changes the expected power.

2. Statistical Analyses and Model Assumptions. The manuscript states that ANCOVA, mixed models, and regression analyses will be performed, but several details are unclear:

o Which covariates will be included in each model?

o Will baseline differences (e.g., cancer treatment type) be adjusted?

o Will interaction effects (group × time) be explicitly tested?

Please provide clear model specifications for ANCOVA and mixed models, including covariates and interaction terms.

3. Missing Data and Attrition Handling. Given the longitudinal design (baseline, 3 months, 6 months), missing data is inevitable. The manuscript does not state how missing values will be handled: Multiple imputation? Last observation carried forward (LOCF)? Sensitivity analyses for missing data patterns? Please clearly describe how missing data will be addressed and what imputation techniques (if any) will be used.

4. Randomization and Blinding Issues. The study uses stratified randomization based on age groups (8–12, 13–17 years), but the manuscript does not specify whether the study is blinded or open-label. If the study is open-label, potential bias in outcome assessment should be discussed, especially for subjective measures like cognitive function. If blinding was not feasible, describe mitigation strategies (e.g., blinded assessors for key outcomes).

5. Interpretation of Biomarker Results

o The study includes immune and inflammatory biomarkers (e.g., IL-6, TNF-alpha) as endpoints.

o However, biological variability in cytokine levels can be high, making interpretation difficult.

o Were pre-specified clinical thresholds for significant biomarker changes defined?

Please discuss how biomarker variability will be handled and whether biological relevance thresholds were considered.

Minor Comments

o Clarify the Hypothesis Testing Strategy, i.e. will separate models be used for each outcome, or will a multivariate approach be applied?

o Address Potential Bias in Patient Selection since the inclusion criterion (1–8 years post-treatment) may introduce survivor bias. Discuss how this may impact the generalizability of findings.

o Ethical and Transparency Considerations, i.e. will raw neuroimaging and biomarker data be made available upon study completion?

Reviewer #2: 1. The introduction effectively outlines the background but lacks a strong justification for why this study is necessary. Consider elaborating on the research gap being addressed.

2. Additional details are required on the selection criteria for participants/samples (if applicable).

3. The statistical analysis methods need more explanation. Ensure all assumptions are stated and justified.

4. Figures and tables should be more clearly labeled and referenced appropriately in the text.

5. Some results are presented without adequate discussion on their implications.

6. Ensure consistency in data reporting and units of measurement.

7. Clarify any contradictory findings and provide a more detailed discussion in relation to previous studies.

8. While the manuscript is generally well-written, there are sections with ambiguous phrasing and grammatical inconsistencies. Consider professional proofreading.

9. The conclusion section should better summarize key findings and their impact.

10. Recommendations for future research should be expanded.

11. Ensure that all references follow the required journal format.

12. Some figures could be improved for clarity and readability.

13. Define all abbreviations when first mentioned.

Reviewer #3: Dear Authors

I have some questions:

1. why do you exam these specific immune and and inflammatory biomarkers?

2. who was included to the control group?

3. who was responsible for neuropsychological evaluation?

4. what is adherence evaluation?

5. the number of patients should be bigger

6. The CNS survivors should be analysed seperately.

**Do you want your identity to be public for this peer review?** For information about this choice, including consent withdrawal, please see our Privacy Policy

Reviewer #1: No

Reviewer #2: No

Reviewer #3: No

---

## [Author Response · Author response to Decision Letter 1]

2 Jul 2025

Dear Academic Editor and Reviewers,

We sincerely thank you for your thorough evaluation of our manuscript entitled "Randomized controlled clinical trial with a multifaceted cognitive training program using video games in childhood cancer survivors." We appreciate the insightful and constructive comments provided by the reviewers and editorial team, which have helped us substantially improve the quality and clarity of the manuscript.

Please find below our point-by-point responses to all concerns raised. For each comment, we have provided the original reviewer/editor request in bold, followed by our response.

Academic editor’s requirements:

The manuscript has been modified to fully comply with PLOS ONE’s formatting and style requirements.

2. Your ethics statement should only appear in the Methods section.

The ethics statement has been moved to the Methods section and removed from all other parts of the manuscript.

3. Please include captions for your Supporting Information files at the end of your manuscript.

Captions for the Supporting Information files are now provided at the end of the manuscript and have been labeled according to the journal’s Supporting Information guidelines. All Supporting Information files have been renamed to correspond with these labels.

4. Remove copyrighted logos from the uploaded protocol.

All copyrighted institutional logos were removed.

Reviewer #1 – Major comments

1. Sample Size Justification and Power Calculation.

The reviewer is correct in noting that G*Power does not support true mixed-effects models. The initial sample size estimation was performed using a fixed-effects model with repeated measures ANCOVA (2 groups × 3 time points), assuming:

- Effect size: f = 0.25 (moderate)

- α = 0.05

- Power (1−β) = 0.95

- Software: G*Power 3.1.9.4

However, we acknowledge that our primary planned analysis involves a mixed-effects model with repeated measures, which allows for modeling intra-subject correlations. The use of a fixed-effects model for power estimation could lead to a slight underestimation of the required sample size, particularly if the intra-class correlation (ICC) is high.

To address this, we have reviewed the potential impact of including intra-subject correlation in the sample size estimation. According to published guidelines and simulations for mixed-models with similar designs, assuming ICC values between 0.3 and 0.6 (common for neurocognitive outcomes), the impact on required sample size remains moderate, particularly given our model with three repeated measures.

For transparency, we have now added a discussion in the revised manuscript noting this limitation and reporting the sample size estimations under different ICC scenarios. For example, assuming ICC = 0.5, and similar parameters, simulations suggest that a total sample size of 42–48 subjects would achieve equivalent power, which we consider feasible, given our planned recruitment.

2. Statistical Analyses and Model Assumptions.

We have now clarified and specified the statistical models in the revised manuscript. In summary:

Mixed ANCOVA (main analysis):

- Between-subjects factor: Group (Intervention vs. Control)

- Within-subjects factor: Time (T0, T+3, T+6)

- Covariates: Age, sex, baseline cognitive scores, cancer diagnosis type, treatment modality (e.g., CNS radiation, chemotherapy with neurotoxic agents)

- Interaction terms: Group × Time interaction is explicitly modeled to assess differential changes across time by group.

Multiple linear regressions (exploratory):

- Predictors: treatment group, age, baseline cognitive profile, psychosocial indicators

- Outcomes: neurocognitive test scores, neuroimaging markers, inflammatory biomarker changes

All models will be checked for assumptions (normality, homoscedasticity, multicollinearity) and complemented with appropriate effect size indices (e.g., omega squared, R², Cohen’s f).

We have added explanations of variable inclusion in the updated Methods section.

3. Missing Data and Attrition Handling.

Thank you for pointing this out. As a longitudinal study, we anticipated some degree of attrition and missing data.

We have now clarified the following:

- Missing data will be handled using Multiple Imputation (MI) under the assumption of missing at random (MAR), which is implemented using chained equations.

- Sensitivity analyses will be conducted by comparing results with and without imputation and using complete-case analyses.

- If patterns suggest non-random missingness, we will perform sensitivity models including inverse probability weighting or pattern-mixture models as appropriate.

We have added a new paragraph in the Statistical Analysis subsection that explicitly describes this.

4. Randomization and Blinding Issues.

The study is partially blinded. Randomization is stratified by age and implemented using REDCap. Clinical follow-up staff will not be blinded, but neuropsychological and neuroimaging assessors will remain blinded to the group allocation to minimize outcome bias.

We have clarified these aspects in the revised “Randomization and Blinding” subsection and discussed the potential for bias and mitigation strategies for unblinded roles.

5. Interpretation of Biomarker Results

We appreciate the reviewer’s thoughtful observation regarding the interpretation of cytokine data. It is indeed well recognized that inflammatory cytokines such as IL-6 and TNF-α exhibit substantial inter- and intra-individual biological variability, influenced by circadian rhythms, age, infection status, physical activity, and psychological stress.

In our study, cytokine levels will be measured at standardized time points and using highly sensitive and validated multiplex immunoassays, in order to minimize pre-analytical variability. Furthermore, the samples will be processed in a centralized laboratory to ensure consistency in handling and assay performance.

Although no universally accepted clinical thresholds exist for defining abnormal cytokine levels in pediatric cancer survivors, we will rely on previously published pediatric reference ranges (de Jager et al., 2009; de Lima et al., 2019) and internal control group distributions to aid interpretation. As such, we did not pre-specify rigid clinical cut-offs, but rather aim to assess within-subject changes over time and between-group differences in the direction and magnitude of the inflammatory response.

Importantly, we plan to complement group-level comparisons with Reliable Change Index (RCI) analyses and regression-based approaches to identify whether individual biomarker changes correlate with neurocognitive outcomes, helping to contextualize biological relevance even in the presence of high variability.

We have added a paragraph in the Discussion section to clarify our strategy for interpreting the biomarker results and the limitations inherent to cytokine variability.

Reviewer #1 – Minor comments

Clarify the Hypothesis Testing Strategy

Each primary outcome (neuropsychological test scores, neuroimaging variables, and biomarker levels) will be analyzed using separate statistical models, given the different data structures and measurement characteristics of each domain. We will not apply a global multivariate test across all outcome types. Within each domain, appropriate multivariate procedures will be considered where applicable (e.g., MANOVA for groups of related cognitive outcomes), but primary hypothesis testing will be conducted through parallel domain-specific analyses.

Address Potential Bias in Patient Selection.

We agree that the inclusion criterion of patients 1–8 years post-treatment may introduce survivor bias. This aspect is now explicitly acknowledged as a limitation in the Discussion section. We have added a note clarifying that our findings may not generalize to all pediatric cancer survivors, particularly those with early relapse or severe neurological sequelae that prevent participation. However, we believe that the selected time window reflects a clinically meaningful survivor population in which cognitive late effects are often the most prominent and intervention is still viable.

Ethical and Transparency Considerations.

The ethical protocol approved by the institutional review board ensures anonymization and secure storage of all patient data. We plan to share de-identified raw datasets, including neuroimaging and biomarker data, upon reasonable request, after the publication of the main findings. This will be noted in the revised “Ethics and dissemination” section. Data sharing will comply with GDPR and institutional policies.

Reviewer #2

1. The introduction lacks a strong justification

To strengthen the rationale for the study, we have added a new paragraph at the end of the Introduction section. This paragraph highlights the current gaps in the literature regarding the variability and unclear incidence of neurocognitive sequelae in pediatric cancer survivors, lack of correlation with neuroimaging findings, and absence of systematic screening protocols or engaging interventions specifically designed for this population. The added paragraph clarifies the unmet clinical and research needs that the present study aims to address.

2. Additional details are required on the selection criteria

We expanded the selection criteria section to include the inclusion and exclusion criteria for the healthy control group. This clarifies the basis for the comparison of neuropsychological and neuroimaging assessments and improves the transparency of our recruitment strategy.

3. The statistical analysis methods need more explanation.

The Statistical Methods section has been thoroughly revised to address this and the comments of other reviewers.

4. Figures and tables should be more clearly labeled and referenced appropriately in the text.

We have re-labeled all figures and tables according to the journal’s guidelines and are appropriately referenced in the text.

5. Some results are presented without adequate discussion on their implications.

We have reviewed the manuscript and expanded the discussion in selected areas to better contextualize the preliminary findings and their potential implications, particularly in relation to neurocognitive outcomes and inflammatory markers.

6. Ensure consistency in data reporting and units of measurement.

We appreciate the reviewer’s comment. We have carefully revised the manuscript to ensure that all results are reported consistently, with standardized units of measurement and terminology across tables, figures, and text.

7. Clarify any contradictory findings and provide a more detailed discussion in relation to previous studies.

We have revised the Discussion section to address the apparent inconsistencies.

8.There are sections with ambiguous phrasing and grammatical inconsistencies

The manuscript has been thoroughly proofread and corrected for its clarity and grammar.

9. The conclusion section should better summarize key findings and their impact.

We have revised the conclusion to summarize the study objectives more clearly and highlight the potential clinical impact of the proposed intervention, as follows:

“Increased survival in pediatric cancer patients highlights the urgent need to diagnose and treat the neuropsychological effects of both the disease and its treatments. Although this condition has been widely studied, we do not have standardized screening protocols or evidence-based cognitive rehabilitation tools currently available in routine clinical practice. By evaluating neuropsychological performance, neuroimaging variables, and blood biomarkers, this study will help to determine the prevalence, severity, and potential mechanisms of neurocognitive impairment in childhood cancer survivors. In parallel, it will assess the feasibility and effectiveness of a multifaceted cognitive training platform using video games, which, if beneficial, could offer a novel, engaging, and scalable strategy for improving long-term outcomes in this population”.

10. Recommendations for future research should be expanded.

Thank you for this suggestion. We have added a brief statement at the end of the Discussion, outlining potential future research directions based on the current study design and objectives, as follows:

“Future studies should explore the long-term sustainability of cognitive improvements following this type of intervention, and examine whether earlier implementation, closer to the end of treatment, may enhance its effectiveness. In addition, further research is needed to validate the relationship between changes in cognitive function and neuroimaging or biomarker profiles in larger multicenter cohorts.”

11. Ensure that all references follow the required journal format.

All references have been revised to comply with the journal guidelines.

12. Some figures could be improved for clarity and readability.

Figure 1 has been enlarged to enhance clarity.

13. Define all abbreviations when first mentioned.

All abbreviations are now defined upon first mention in the text.

Reviewer #3

1. why do you exam these specific immune and and inflammatory biomarkers?

Thank you for this question. The selected cytokines (IL-2, IL-4, IL-6, IL-10, IL-17a, TNF-α, IFN-γ, and IL-1 receptor antagonist), along with lymphocyte subpopulations, were chosen based on prior evidence linking immune dysregulation and systemic inflammation to cancer-related cognitive impairment (CRCI). In particular, IL-6 and TNF-α have been repeatedly implicated in neuroinflammation and cognitive dysfunction in both cancer and non-cancer populations (Cheung et al., 2015; Janelsins et al., 2014).

IL-10 and IL-1RA are anti-inflammatory markers that may reflect compensatory or regulatory immune responses. The inclusion of Th1, Th2, and Th17 cytokines provides a broader immunological profile that may help elucidate the inflammatory balance in pediatric cancer survivors.

This panel also aligns with the cytokines evaluated in previous studies of cognitive outcomes in pediatric and adult oncology, supporting comparability and hypothesis-driven interpretation.

We have added a paragraph in the Methods section explaining this.

2. who was included to the control group?

The selection criteria section has been completed to explain the inclusion of patients in the control group.

3. who was responsible for neuropsychological evaluation?

Neuropsychological evaluations will be performed by clinical neuropsychologists from the Department of Psychiatry, Clinical Psychology, and Mental Health of the hospital. This information has been included in the “Variable and instruments” subsection of the manuscript.

4. what is adherence evaluation?

Table 1 shows the different times of the study in which adherence to treatment will be verified. This will be performed using the systems included in some of the gaming platforms, and by telephone/video calls to the participants. This information is detailed now in table.

5. the number of patients should be bigger

The sample Size Justification and Power Calculation have been revised and completed in the Statistical Methods section regarding all reviewers’ comments.

6. The CNS survivors should be analysed seperately.

We agree with the reviewer that patients with CNS involvement may present with distinct neurocognitive profiles and should be carefully considered. In our analysis plan, we have specified that the diagnostic group (CNS vs. non-CNS tumors) will be included as a covariate in all primary models. Additionally, exploratory subgroup analyses will be performed to compare the cognitive and neuroimaging outcomes between CNS and non-CNS survivors, where the sample size allows. We have clarified this in the revised Statistical Methods section.

We are grateful for the reviewers’ thoughtful feedback and believe that the revised manuscript has been substantially improved. We hope that the revised version meets the journal’s expectations, and thank you for your time and consideration.

Sincerely,

Antonio Perez-Martinez

---

## [Decision Letter · Decision Letter 1]

23 Jul 2025

Study protocol for a randomized controlled clinical trial of a multifaceted cognitive training program using video games in childhood cancer survivors

PONE-D-24-47925R1

Dear Dr. González-Pérez,

We’re pleased to inform you that your manuscript has been judged scientifically suitable for publication and will be formally accepted for publication once it meets all outstanding technical requirements.

Kind regards,

Miray Budak

Academic Editor

PLOS ONE

Additional Editor Comments (optional):

Reviewers' comments:

Reviewer's Responses to Questions

Comments to the Author

1. Does the manuscript provide a valid rationale for the proposed study, with clearly identified and justified research questions?

Reviewer #1: Yes

2. Is the protocol technically sound and planned in a manner that will lead to a meaningful outcome and allow testing the stated hypotheses?

Reviewer #1: Yes

3. Is the methodology feasible and described in sufficient detail to allow the work to be replicable?

Reviewer #1: Yes

4. Have the authors described where all data underlying the findings will be made available when the study is complete?

Reviewer #1: Yes

5. Is the manuscript presented in an intelligible fashion and written in standard English?

Reviewer #1: Yes

You may also provide optional suggestions and comments to authors that they might find helpful in planning their study.

Reviewer #1: Thank you for addressing the comments.

All major concerns were addressed methodically. Revisions improved transparency (e.g., power analysis, biomarker interpretation).

**Do you want your identity to be public for this peer review?** For information about this choice, including consent withdrawal, please see our Privacy Policy

Reviewer #1: No

---

## [Editor Report · Acceptance letter]

PONE-D-24-47925R1

PLOS ONE

Dear Dr. González-Pérez,

I'm pleased to inform you that your manuscript has been deemed suitable for publication in PLOS ONE. Congratulations! Your manuscript is now being handed over to our production team.

Kind regards,

on behalf of

Dr. Miray Budak

Academic Editor

PLOS ONE